# The Class I HDAC Inhibitor, MS-275, Prevents Oxaliplatin-Induced Chronic Neuropathy and Potentiates Its Antiproliferative Activity in Mice

**DOI:** 10.3390/ijms23010098

**Published:** 2021-12-22

**Authors:** Sylvain Lamoine, Mélissa Cumenal, David A. Barriere, Vanessa Pereira, Mathilde Fereyrolles, Laëtitia Prival, Julie Barbier, Ludivine Boudieu, Emilie Brasset, Benjamin Bertin, Yoan Renaud, Elisabeth Miot-Noirault, Marie-Ange Civiale, David Balayssac, Youssef Aissouni, Alain Eschalier, Jérôme Busserolles

**Affiliations:** 1UMR 1107 Inserm/UCA, CHU Clermont-Ferrand, Université Clermont Auvergne, Neuro-Dol, 63000 Clermont-Ferrand, France; sylvain.lamoine@uca.fr (S.L.); melissa.cumenal@ucalgary.ca (M.C.); david.a.barriere@gmail.com (D.A.B.); vanesp_63@hotmail.com (V.P.); mathilde.fereyrolles@ans-biotech.com (M.F.); laetitia.prival@uca.fr (L.P.); julie.barbier@uca.fr (J.B.); lboudieu@chu-clermontferrand.fr (L.B.); david.balayssac@uca.fr (D.B.); youssef.aissouni@inserm.fr (Y.A.); alain.eschalier@uca.fr (A.E.); 2iGReD, CNRS, INSERM, Faculté de Médecine, Université Clermont Auvergne, 63000 Clermont-Ferrand, France; emilie.brasset@uca.fr (E.B.); benjamin.bertin@uca.fr (B.B.); yoan.renaud@uca.fr (Y.R.); 3UMR 1240 INSERM IMoST, Université Clermont Auvergne, 58 Rue Montalembert, 63000 Clermont-Ferrand, France; elisabeth.noirault@uca.fr; 4ACCePPT—AutomédiCation aCcompagnement Pluriprofessionnel PatienT, Université Clermont Auvergne, 63000 Clermont-Ferrand, France; marie-ange.civiale@uca.fr; 5CHU Clermont-Ferrand, Délégation à la Recherche Clinique et à l’Innovation, 63000 Clermont-Ferrand, France; 6Institut Analgesia, Faculté de Médecine, BP38, 63001 Clermont-Ferrand, France

**Keywords:** colorectal cancer, oxaliplatin, peripheral neuropathy, histone deacetylase inhibitor, MS-275, *APC^Min/+^* mice, T84 cells, HT29 cells, CT26 cells, orthotopic model of colorectal cancer

## Abstract

Oxaliplatin, the first-line chemotherapeutic agent against colorectal cancer (CRC), induces peripheral neuropathies, which can lead to dose limitation and treatment discontinuation. Downregulation of potassium channels, which involves histone deacetylase (HDAC) activity, has been identified as an important tuner of acute oxaliplatin-induced hypersensitivity. MS-275, a class I histone deacetylase inhibitor (HDACi), prevents acute oxaliplatin-induced peripheral neuropathy (OIPN). Moreover, MS-275 exerts anti-tumor activity in several types of cancers, including CRC. We thus hypothesized that MS-275 could exert both a preventive effect against OIPN and potentially a synergistic effect combined with oxaliplatin against CRC development. We first used RNAseq to assess transcriptional changes occurring in DRG neurons from mice treated by repeated injection of oxaliplatin. Moreover, we assessed the effects of MS-275 on chronic oxaliplatin-induced peripheral neuropathy development in vivo on *APC^Min/+^* mice and on cancer progression when combined with oxaliplatin, both in vivo on *APC^Min/+^* mice and in a mouse model of an orthotopic allograft of the CT26 cell line as well as in vitro in T84 and HT29 human CRC cell lines. We found 741 differentially expressed genes (DEGs) between oxaliplatin- and vehicle-treated animals. While acute OIPN is known as a channelopathy involving HDAC activity, chronic OIPN exerts weak ion channel transcriptional changes and no HDAC expression changes in peripheral neurons from OIPN mice. However, MS-275 prevents the development of sensory neuropathic symptoms induced by repeated oxaliplatin administration in *APC^Min/+^* mice. Moreover, combined with oxaliplatin, MS-275 also exerts synergistic antiproliferative and increased survival effects in CT26-bearing mice. Consistently, combined drug associations exert synergic apoptotic and cell death effects in both T84 and HT29 human CRC cell lines. Our results strongly suggest combining oxaliplatin and MS-275 administration in CRC patients in order to potentiate the antiproliferative action of chemotherapy, while preventing its neurotoxic effect.

## 1. Introduction

Oxaliplatin, in combination with 5-fluorouracil, is a standard treatment option for primary and metastasized colorectal cancer [1], which induces a peripheral neuropathy, known to be uniquely due to oxaliplatin [2], which can lead to dose limitation and treatment discontinuation [2,3]. Oxaliplatin-induced peripheral neuropathy (OIPN) is characterized by paresthesias and dysesthesias, which can be triggered or exacerbated by cold exposure as soon as after the first infusion of the drug in 90% of patients [2,4]. These neuropathic symptoms do not always completely resolve between treatment cycles [2], and 30–50% of patients suffer from chronic OIPN [5]. Currently, none of the drugs used for prevention or treatment of OIPN have been shown to be sufficiently effective to be routinely incorporated into clinical practice [6,7], and strategies to control neuropathy rely essentially on modifications of dosage and infusion scheme, thus decreasing the chances of survival [8]. Moreover, cumulative doses of chemotherapy are probably not a predictive factor of chemotherapy-induced neuropathy (CIPN) [9], suggesting that modifications of dosage and infusion scheme might not affect the development of neuropathic symptoms in cancer survivors.

Evidence has shown that oxaliplatin causes ion channel expression modulations in dorsal root ganglia (DRG) neurons [10,11,12,13,14,15,16], which are thought to contribute to acute peripheral hypersensitivity. Most dysregulated genes encode for ion channels involved in cold and mechanical perception, including members of the transient receptor potential (TRP), the two-pore potassium channels (K2P) and the voltage-dependent potassium (Kv) families. We recently demonstrated that oxaliplatin-mediated downregulation of K^+^ channels of the K2P and Kv families involves a transcription factor known as the neuron-restrictive silencer factor (NRSF), and its epigenetic corepressors, class I histone deacetylases (HDACs) [17]. Furthermore, the class I HDAC inhibitor, MS-275, exerted a preventive effect against both neuropathy and K^+^ channels downregulation after a single-dose of administered oxaliplatin [17]. Whether sustained peripheral epigenetic events are involved in OIPN chronicization is not known. Oxaliplatin-induced chronic neuropathy is rather thought to be caused by platinum accumulation in DRG neurons [18,19,20], leading to morphologic and functional changes in DRG neurons. Mitochondrial toxicity, as well as satellite glial cell activation in DRG, favoring the development of neuroinflammation, could also contribute to chronic OIPN establishment [21,22]. Besides—or as a consequence of peripheral sensitization—cumulative evidence suggests that spinal neuronal plasticity plays a critical role in the persistence of OIPN [23,24]; with the mechanisms still poorly defined. In the present study, we used repeated oxaliplatin administration and analyzed whole transcriptomic changes that occur in DRG neurons from OIPN mice as compared with vehicle-treated animals.

Since we observed a preventive effect of MS-275 against acute OIPN development, we wondered whether this molecule would be able to exert a preventive effect against chronic OIPN. Furthermore, HDAC inhibitors are promising therapeutic molecules against several pathological states, including cancer [25]. Several class I HDACs—HDAC1, 2, 3—and eight isotypes have been involved in colorectal cancer [26,27]. In a recent meta-analysis, the expression level of HDAC1 in colorectal cancer was found to be higher and closely associated with tumor stage and tumor grade than that in noncancerous tissue. In addition, patients with low HDAC1 expression showed better overall survival than did those with high HDAC1 expression in gastrointestinal malignancy [26]. HDAC3, another class I HDAC isotype, was found to be upregulated in 52.1% of colorectal tumor tissue specimens [28].

In this context, we hypothesized that MS-275 could exert both a preventive effect against OIPN and potentially a synergistic effect combined with oxaliplatin against CRC development. Thus, we assessed the effects of MS-275 on chronic OIPN development and on cancer progression when combined to oxaliplatin both in vitro on different human colorectal cancer cell lines and in vivo on mice models of colorectal cancer.

## 2. Results

### 2.1. Transcriptomic Changes Observed in DRG Neurons from Mice Treated with Repeated Oxaliplatin Administration

Prolonged oxaliplatin treatment is known to induce persistent peripheral neuropathy that particularly affects extremities. We evaluated the development of these symptoms in C57Bl/6J mice administered with a clinically relevant dose of oxaliplatin (3 mg/kg, i.p.) twice per week for 3 weeks (Figure 1A). Cold and mechanical hypersensitivity develop from day 4 and persisted until the end of the experiment (day 21) in oxaliplatin-treated mice, as evaluated using the paw immersion test (Figure 1B, Appendix A) and von Frey test, respectively (Figure 1C, Appendix A).

We then performed a comparative RNAseq analysis in DRG neurons of oxaliplatin- and vehicle-treated animals at end-point of the experiment. We found 741 differentially expressed genes (DEGs) between oxaliplatin- and vehicle-treated animals. Respectively, 342 and 399 genes were up- and downregulated in OIPN DRG neurons as compared to control animals (Figure 1D,E. For full list of upregulated and downregulated genes following oxaliplatin treatment, see Supplementary data S1 and S2, respectively). Of note, while acute OIPN is known as a channelopathy, including transcriptional variation of a variety of ion channels involved in cold and pain perception, only a discrete number of ion channels (GIRK1 and Kv3.3) showed distinct mRNA levels in DRG neurons between OIPN and sham animals at day 21. We previously showed an increase in HDAC3 expression in DRG neurons from mice administered with a single dose of oxaliplatin [17]. However, in the present work, we failed to observe any transcriptional differences of class I HDACs (HDAC1, HDAC2, and HDAC3) in mice treated with repeated dose of vehicle or oxaliplatin. However, several genes encoding for methyl transferase (Setd1b, Mettl8, kmt2d) were found to be downregulated, and genes associated to histone deacetylase complex such as Sap30l [29] and Banp [30] were upregulated in DRG neurons from oxaliplatin-treated animals (see supplementary data S1). As DRGs are a heterogeneous tissue containing the cell bodies of peripheral nerves, epithelial cells, fibroblasts, glial cells, and immune cells, DEGs discovered following treatment with oxaliplatin cannot be unequivocally assigned to sensory neurons alone. We thus performed CTEN, a web-based analytical platform using a highly expressed, cell-specific gene database to identify enriched cell types from transcriptomic data [31] to gain insights into cell-specific effects of oxaliplatin (Figure 2A). We also performed GO-enrichment analysis to further elucidate potential biological functions (Figure 2B) and cellular components (Figure 2C) associated with the 741 overlapping DEGs. For molecular functions, the overlapping DEGs were mainly associated with axogenesis, axon development, myelin sheath, and actin and filament organization. Cellular organelles (Golgi, endoplasmic reticulum, and mitochondria) component genes were also significantly differentially expressed.

### 2.2. MS-275 Prevents Oxaliplatin-Induced Chronic Neuropathy in Mice

We recently showed that the class I HDAC inhibitor, MS-275, significantly prevented pain symptoms induced after a single oxaliplatin injection in mice [17]. Moreover, it is increasingly recognized that histone modifications play a crucial role in cancer initiation and progression, including in CRC [32]. In vitro experiments and animal models have shown anti-tumor activities of HDAC inhibitors [33]. Thus, we hypothesized that MS-275 would potentially exert beneficial effects, both against neuropathic symptoms induced by oxaliplatin and against colorectal cancer progression. For that, we evaluated the development of these symptoms in mice administered with oxaliplatin (3 mg/kg, i.p.) twice per week for 3 weeks with or without MS-275 (15mg/kg) administered by oral gavage half an hour before each oxaliplatin injection (Figure 3A). We first assessed the effect of oxaliplatin and MS-275 on *APC^Min/+^* mice, a model of intestinal neoplasia, and on the C57BL/6J background [34]. Cold hypersensitivity developed in oxaliplatin-treated *APC^Min/+^* mice (Figure 3B, Appendix A) from day 4 after the first injection as evidenced by the significant decrease in tail withdrawal latency in response to a noxious cold stimulation (10 °C). Mechanical hypersensitivity was also observed with a similar time course in oxaliplatin-treated animals (Figure 3C, Appendix A). MS-275 administration significantly prevented both cold (Figure 3B) (two-way ANOVA analysis: treatment effect (*p* < 0.0001); time effect (*p* = 0.0004), interaction (*p* = 0.0174)) and mechanical hypersensitivities (Figure 3C) (two-way ANOVA analysis: oxaliplatin effect (two-way ANOVA analysis: treatment effect (*p* = 0.0002); MS-275 effect (*p* < 0.0001), interaction (*p* = 0.0035)) induced by oxaliplatin.

### 2.3. General and Hematologic Toxicity Profile of Oxaliplatin and MS-275 Combination

The body weight of animals was monitored twice per week during all experimental procedures. Oxaliplatin slowed down body weight gain in *APC^Min/+^*. Oxaliplatin-treated animals had significant lower body weight at end point (D21). On the contrary, MS-275 did not affect this parameter and did not worsen this oxaliplatin side effect (Figure 3D, Appendix A) (two-way ANOVA analysis: treatment effect (*p* = 0.0143); time effect (*p* < 0.0001), interaction (*p* < 0.0001).

Hematologic toxicity represents a known adverse side effect of several anticancer chemotherapeutic agents. Consistently, we observed a significant erythropenia at end point (day 21 after the first infusion) in oxaliplatin-treated animals as compared with control animals (Figure 3E, Appendix A). Noteworthy, MS-275 did not demonstrate any effect on red blood cells and had no additional effect when combined with oxaliplatin (Figure 3E) (two-way ANOVA analysis: oxaliplatin effect (*p* < 0.0001); MS-275 effect (*p* = 0.7926), interaction (*p* = 0.0805)). On the contrary, MS-275 is known to exert lymphocytopenia, neutropenia and thrombocytopenia [35]. In our experimental conditions, MS-275 significantly decreased white blood cells counts in *APC^Min/+^* as compared with control mice (Figure 3F, Appendix A). Oxaliplatin did not exert any effect on white blood cells and did not have any synergic effect with MS-275 on this parameter (Figure 3F) (two-way ANOVA analysis: oxaliplatin effect (*p* = 0.5441); MS-275 effect (*p* < 0.0001), interaction (*p* = 0.6229)). None of the drugs, whether used alone or in combination, affected the number of thrombocytes (Figure 3G, Appendix A) (two-way ANOVA analysis: oxaliplatin effect (*p* = 0.5638); MS-275 effect (*p* = 0.6911), interaction (*p* = 0.9272)). These results suggest that co-administration of MS-275 and oxaliplatin would have additive, but not synergic, hematological toxicities that should be taken into consideration since in chemotherapy- treated patients, white blood cell count is an important marker for the continuation of the treatment.

### 2.4. MS-275 and Oxaliplatin Antiproliferative Effects in Familial Adenomatous Polyposis (FAP) Mice and in CT26 Tumor-Bearing Mice

*APC^Min/+^* mice harbor a germline mutation in the *Apc* gene at codon 850 that leads to a truncation of the protein [34]. In *Apc**^Min/+^* mice, adenomas begin to develop in infancy, with death occurring at 16–20 weeks due to chronic intestinal hemorrhage [36]. We assessed the effect of the drugs on the number of polyps in *APC^Min/+^* mice at necropsy. The dose of oxaliplatin used (3 mg/kg, i.p.), while inducing strong neuropathic adverse effects, failed to impact polyposis in this model of FAP (Figure 3H, Appendix A). MS-275 also failed to significantly decrease the number of polyps in *APC^Min/+^* as well, but a significant interaction was detected using the two-way ANOVA analysis (two-way ANOVA analysis: oxaliplatin effect (*p* = 0.2037); MS-275 effect (*p* = 0.6351), interaction (*p* = 0.0443)), suggesting a synergistic effect of combined drugs treatment in this model.

Additionally, we used a mouse model of an orthotopic allograft of the CT26 cell line (mouse colorectal cancer cells) that was stably transfected with the luciferase gene. This model consists of grafting a piece of tumor onto the caecum of animals [37] and to follow tumor progression using bioluminescence (Figure 4A). As shown in Figure 4B (Appendix A), animals receiving the vehicle treatment show a rapid tumor growth that is hampered in the mice treated with oxaliplatin from day 7 after the start of treatment. MS-275, either alone or combined with oxaliplatin, also strongly slowed cancer progression in this model (two-way repeated measure ANOVA analysis: treatment effect (*p* = 0.0005); time effect (*p* < 0.0001), interaction (*p* < 0.0001)). Bioluminescence results are shown until D10 because of the occurrence of death events at D12 in the vehicle group, in which 100% animals died before D23 (Figure 4C, Appendix A). Oxaliplatin improves the survival of the mice since 50% of the treated animals are still alive at the end of the treatment and 30% at the end of the follow-up on D40 (*p* = 0.0064). MS-275 alone also significantly improves mice survival since 66% of animals were still alive after end of treatment and 50% at D40 (*p* = 0.0019). Drugs association demonstrated an increased benefit for survival, as shown by their synergistic effect on mean and median survival (Figure 4 and Appendix A).

### 2.5. MS-275 and Oxaliplatin Effects on Human Cancer Cell Viability

The combined effect of oxaliplatin and MS-275 was monitored on the viability of two human colon cancer cell lines, T84 and HT29, using the MTT test. Oxaliplatin (1 to 64µM) exerted a dose-dependent effect, decreasing both cell line viabilities (Figure 5A,B). MS-275 (1 to 25 µM) also significantly decreased HT29 viability in a dose-dependent manner (Figure 5B, Appendix A). In contrast, it failed to affect T84 cell viability, except at the highest dose tested (25 µM) (Figure 5A, Appendix A). A recent report found large variation in maximal plasma concentrations from 4 to 53.1 ± 92.4 µM after MS-275 administration in humans [38], whereas therapeutically achievable concentrations of oxaliplatin have been reported to be around 5–10 µM [39]. For subsequent analysis, we chose concentrations in the upper range, but these are still therapeutically achievable concentrations for both drugs. The combined effect of MS-275 (2.5 µM) and oxaliplatin (16 µM) was assessed on T84 and HT29 cell viability (Figure 5C,D). In T84 cells, two-way ANOVA analysis made on cell viability data reveals an oxaliplatin effect (*p* < 0.0001), MS-275 effect (*p* = 0.9492), interaction (*p* = 0.1667)) (Figure 5C, Appendix A). In HT29 cells, two-way ANOVA analysis made on cell viability data reveals an oxaliplatin effect (*p* < 0.0001), MS-275 effect (*p* < 0.0001), and interaction (*p* = 0.0001)), (Figure 5D, Appendix A).

### 2.6. MS-275 and Oxaliplatin Effects on Human Cancer Cells Cycle

To investigate the cytostatic effect of both drugs, cell cycle analysis was performed using PI incorporation, followed by flow cytometry analysis (Figure 5E,F). Oxaliplatin exerted an S-phase delay accompanied by a G0/G1 phase reduction on both T84 and HT29 cell lines. As previously shown on the HCT116 cell line, MS-275 induced a G0/G1 blockade and a strong S-phase reduction at a dose of 2.5 µM in both cell lines (Figure 5E,F). Combined treatment resulted in a significant reduction of G0/G1 phase in T84 cells (Figure 5E,Appendix A), while it significantly decreased the S phase and increased the G2/M phase in HT29 cells (Figure 5F, Appendix A). It is known that the cytostatic action of oxaliplatin is mediated through a G2/M blockade in T84 and HT29 cell lines after 72 h of treatment [20]. In our case, only 48 h of treatment were completed, and this earlier stage could explain the S-phase delay observed after oxaliplatin treatment. However, the addition of MS-275 to oxaliplatin seems to accelerate the oxaliplatin cytostatic effect, resulting in a G2/M blockade on HT29 cells (Figure 5F).

### 2.7. Oxaliplatin and MS-275 Act Synergistically to Induce Apoptosis In Vitro

Oxaliplatin is known to disrupt cell proliferation by induction of apoptosis and cell cycle inhibition, through DNA adduct formation [40]. In order to assess the effects of oxaliplatin and MS-275 to trigger apoptosis and cell death, Annexin V and propidium iodide (PI) staining were performed on T84 and HT29 cell lines, followed by flow cytometry analysis (Figure 6). In T84 cells, oxaliplatin (16µM) significantly induced apoptosis, while MS-275 (2.5 µM) had no effect (two-way ANOVA analysis: oxaliplatin effect (*p* = 0.0004); MS-275 effect (*p* = 0.5484), interaction (*p* = 0.9226)) (Figure 6B, Appendix A). When MS-275 was used at higher concentration (12.5 µM), it significantly induced apoptosis and exerted a synergic effect with oxaliplatin on this parameter (two-way ANOVA analysis: oxaliplatin effect (*p* = 0.0003); MS-275 effect (*p* = 0.0002), interaction (*p* = 0.6639)) (Figure 1C, Appendix A). Oxaliplatin (16 µM) and MS-275, only at the highest dose (12.5 µM), induced a significant cell death effect in T84 cells. A synergic effect was observed on this parameter when oxaliplatin (16 µM) was co-administered with MS-275 (2.5 µM) (two-way ANOVA analysis: oxaliplatin effect (*p* = 0.0004); MS-275 effect (P < 0.0001), interaction (*p* = 0.3297)) (Figure 6C, Appendix A) or with MS-275 (12.5 µM) (two-way ANOVA analysis: oxaliplatin effect (*p* = 0.0004); MS-275 effect (*p* < 0.0001), interaction (*p* = 0.3297)) (Figure 1E, Appendix A). In the HT29 cell line, oxaliplatin (16 µM) and MS-275 (2.5 µM) had a weak apoptosis inducing effect but demonstrated a synergic effect (two-way ANOVA analysis: oxaliplatin effect (*p* = 0.0020); MS-275 effect (*p* = 0.0002), interaction (*p* = 0.3784)) (Figure 6E). The same pattern was observed when oxaliplatin (16 µM) was co-administered with a higher MS-275 concentration (12.5 µM) (two-way ANOVA analysis: oxaliplatin effect (*p* = 0.2046); MS-275 effect (*p* = 0.0077), interaction (*p* = 0.8588)). Both oxaliplatin and MS-275 significantly induce cell death, and we found a synergic effect of these drugs on this parameter in HT29 cells either with MS-275 at 2.5 µM ((two-way ANOVA analysis: oxaliplatin effect (*p* < 0.0001); MS-275 effect (*p* < 0.0001), interaction (*p* = 0.0027)) (Figure 6F, Appendix A) or with higher MS-275 concentration (12.5µM) (two-way ANOVA analysis: oxaliplatin effect (*p* = 0.0063); MS-275 effect (*p* < 0.0001), interaction (*p* = 0.3287)) (Appendix AF, Appendix A).

## 3. Discussion

The results obtained in the present study demonstrate that the class I HDAC inhibitor MS-275 prevents the development of sensory neuropathic symptoms induced by repeated oxaliplatin administration in mice. MS-275 also exerts antiproliferative effects in defined in vitro and in vivo CRC models.

While acute neuropathic pain symptoms induced by oxaliplatin-based chemotherapy could be due to changes in ion channel gene expression in DRG neurons, chronic neurotoxicity that occurs with cumulative dosing of oxaliplatin is rather known to involve nuclear DNA damage, mitochondrial damage, overload oxidative stress, glia activation and neuroinflammation [41]. The transcriptomic analysis that we performed on DRG neurons from C57Bl/6J mice treated with repeated oxaliplatin administration is more likely in line with the results obtained by Starobova et al. [42] since oxaliplatin treatment affected mainly neuronal genes in our conditions. No gene coding for depolarizing ion channels was increased and only very few potassium channels–encoding genes were downregulated in chronic OIPN mice DRG neurons, among them being Kv3.3. Oxaliplatin-mediated downregulation of K^+^ channels of the K2P and Kv families we previously observed [17] should thus be transient. On the contrary, while Kv3.3 mRNA was not decreased in acute OIPN animals [17], this K^+^ channel has recently been shown to be downregulated in DRG neurons from oxaliplatin-treated APC^Pirc/+^ rats after repeated oxaliplatin treatment cessation [43]. Our results are thus in line with and suggest the involvement of Kv3.3 downregulation in sustained occurrence of sensory symptoms induced by repeated oxaliplatin injections. In addition, no overexpression of NRSF and HDAC3, or epigenetic factors involved in K^+^ channel downregulation [17], was observed in DRG neurons from chronic oxaliplatin-treated animals. These results suggest that HDAC inhibition might not be effective once OIPN has set in. MS-275 has already been shown to prevent the development of persistent mechanical hypersensitivity after trigeminal nerve injury [44] and to produce analgesia in sciatic nerve injury models [45]. In these later conditions, drug treatment did not affect already-established neuropathic pain after spinal nerve transection, suggesting that histone acetylation might be specifically involved in the emergence of hypersensitivity. Since MS-275 significantly prevented oxaliplatin-induced acute neuropathy [17], we hypothesized that this molecule could also act as a preventive therapeutic drug in a “programmed” chronic oxaliplatin-induced neuropathic condition. We effectively demonstrate that MS-275 prevents oxaliplatin-induced chronic sensory neuropathic symptoms development in *APC^Min/+^* mice. Denk et al. [45] stated that MS-275 likely exerted its effect centrally within the spinal cord in models of traumatic nerve injury and antiretroviral drug-induced peripheral neuropathy. Our recently published data suggest that MS-275 acts by preventing peripheral sensitization induced by a single dose of oxaliplatin [17]. In the present study, MS-275 was administered using a systemic route. We suspect a peripheral action as a key element in the prevention of chronic OIPN development. However, we cannot rule out an additional central effect of the drug.

Safety of anticancer drug associations represent a matter of concern. The hematologic toxicity of oxaliplatin was reported to be mild to moderate, even when associated with 5FU-FA [46]. After oxaliplatin infusion, up to 40% of blood platinum is found in erythrocytes [47], where it is able to form adducts with hemoglobin that may be associated with toxic effects, such as anemia [48]. We observed an important and significant drop of erythrocytes after three weeks of oxaliplatin treatment at the cumulative dose of 18 mg/kg in *APC^Min/+^* mice. This drug was reported to significantly affect RBC count at a higher cumulative dose (30 mg/kg) in C57Bl/6J mice [49]. This suggests that, as shown for neuropathy [43], hematological toxicity may be enhanced in cancer conditions, and it reinforces the need for preclinical studies in experimental models of cancer. MS-275 is known to induce a dose-dependent decrease of thrombocytes and leucocytes both in human [50] and in rodents [51]. In our experimental conditions, MS-275 did not affect RBC nor thrombocytes count, but it significantly decreased white blood cells counts in *APC^Min/+^* mice. Of note, the association of MS-275 with oxaliplatin did not worsen the hematological toxicities induced by individual drugs, and MS-275 did not increase oxaliplatin-induced reduced weight gain. The overall safety of combined drug administration seems acceptable, suggesting a possible association of MS-275 with oxaliplatin in humans to prevent oxaliplatin-induced neuropathy, but close monitoring will be needed regarding the hematological impact of individual drugs.

The next question was to evaluate the combined effects of MS-275 and oxaliplatin on CRC development. While oxaliplatin, at a dose that induces neuropathic symptoms, and MS-275 did not individually significantly decreased the total number of polyps in *APC^Min/+^* mice, we observed a significant synergistic effect of oxaliplatin and MS-275 co-administration. It is known that loss of APC function leads to increased expression of HDAC2 in intestinal epithelial cells and tumors [52]. Inhibition of the enzymatic activity of class I HDACs and induction of HDAC2 degradation with valproic acid has previously been shown to reduce the number and sizes of intestinal adenomas in *APC^Min/+^* mice [52]. We hypothesized that the lack of effect of MS-275 in our conditions could rely on the dose used, and/or on the different inhibitory profile of valproic acid and MS-275 on the different HDAC isoforms. Using the same dose regimen, both oxaliplatin and MS-275 exert antiproliferative effects, and combined-drug administration significantly impedes tumor progression and increases overall survival in an orthotopic graft tumor-bearing mouse model. These results are consistent with recent data obtained in vitro and in vivo in a transplant study in which female BALB/c mice received subcutaneous injection of CT26 cells [53]. CT26 was derived from an undifferentiated colorectal carcinoma induced in a BALB/c mouse by repeated intrarectal instillations of N-nitroso-N-methylurethan and was shown to be modestly immunogenic [54]. In addition to acting on tumor cells, MS-275 also acted on host cells in the immune system [55], which could account for additional benefit when combined with oxaliplatin in this model. When tested on human CRC cells, both drugs, individually, demonstrated different antiproliferative profile depending on the cell line. Although efficient in both T84 and HT29 cells, oxaliplatin exerted improved efficacy in the T84 cell line. Organic cation transporter 3 (OCT3) is an interesting target for oxaliplatin resistance because of its implication on oxaliplatin transport and efficacy [56]. The expression of OCT3, which is known to be weaker in HT29 than in T84 cells, has been shown to be correlated with platinum accumulation in these cells and with oxaliplatin cytotoxicity [56], which can sustain our results. MS-275 has previously been shown to be less effective in p53 null cells [57]. Consistently, we observed that T84 cells, that do not express p53 [58] demonstrated a strong resistance, whereas HT29, which expresses p53, was highly sensitive toward the antiproliferative effect of MS-275. HT29 was previously shown to be a responder to MS-275, which might increase the adhesive properties of these cells, thus preventing their metastatic spread and immune escape [59]. On the contrary, T84 cells can thus be considered as a weak or non-responder, as it has been shown for other CRC cell lines for which MS-275 induced downregulation of genes involved in cell adhesion [59]. Whether such genes regulation also occurs in T84 cells will be of interest in order to define biomarkers to predict the response to MS-275. Whatever the sensitivity of these cell lines toward oxaliplatin and MS-275, combined drug associations result in synergistic apoptotic and cell death effects. These results are concordant with those obtained in HCT116 cells in which MS-275 has been shown to enhance apoptosis induced by oxaliplatin [53]. At the highest doses tested, apoptosis accounts for more than 67% and 86% of the cell death effect of MS-275 solely or MS-275 and oxaliplatin combined, respectively in T84 cells, while it accounts for only 30% and 31% of these effects in HT29 cells. Of note, MS-275 was proposed to promote mainly caspase-dependent cell death in p53+ cells, whereas, in p53- cells, caspase inhibition switches the mode of cell death to a mainly non-caspase-dependent one [57]. HDACi are epigenetic drugs that enhance protein acetylation and thereby influence a large number of cellular functions. Global assays and analyses revealed that HDACi promote an induction of replicative stress and DNA damage [60], and that they disrupt functional EMT/MET protein expression signatures and trigger apoptosis of cultured cancer cells [61]. MS-275 induced autophagy has also been involved in human colon cancer cell death (HCT116) [62], and could thus be differently involved in both cell lines in our conditions. It is increasingly recognized that HDACi sensitize tumor cells to chemotherapeutics, causing replication stress and DNA damage [8,9,10]. Combined application of irinotecan and entinostat synergistically kills CRC cells in vitro and in vivo, and the induction of acetyl-p53-BAK complexes inducing MOMP upon RS C-terminal hyperacetylation of p53 is an absolute requirement for apoptosis induction by irinotecan and entinostat [63]. More work is needed to understand the mechanisms involved in the beneficial effects observed with MS-275 and oxaliplatin association in our conditions. Future work will include experiments helping to obtain more information on the potential of both drugs to induce replicative stress and DNA damage at different time points.

## 4. Methods

### 4.1. Animals and Models

Procedures were evaluated by a regional ethics committee (CEMEA Auvergne) before approval by the French Ministry of Research and Education (project N° APAFIS#21980) under the European 2010/63/ UE directive. Animal studies are reported in compliance with the ARRIVE 2.0 guidelines [64]. All efforts were taken at each stage of the experiments to limit the numbers of animals used and any discomfort to which they might be exposed, especially in pain experiments.

Experiments were performed using 20-25g C57BL/6J and *Apc^Min/+^* male mice and female BALBc/AnN mice. *Apc^Min/+^* mice were a gift from Dr. Mathilde Bonnet (UMR U1071 Inserm, Clermont-Ferrand, FRANCE). All mice were housed in grouped cages in a temperature-controlled environment with food and water ad libitum. Behavioral experiments were conducted in a quiet room, blind to the treatment, by the same experimenter taking care to avoid or minimize any discomfort of the animals.

For the mouse allograft colorectal cancer model, 5 female BALBc/AnN mice under isoflurane anesthesia were subcutaneously injected in their right flank with 0.5x10^6^ CT26 cells, stably transfected with the Luciferase gene. Mice were housed and monitored until tumors reached a sufficient size (1000–2000 mm^2^) to perform grafts. Mice were then sacrificed, tumors were sampled, and 25 mm^2^ grafts were sized up in DMEM medium on ice. “Recipient mice” were incised along 1 cm in the lower abdomen, under isoflurane anesthesia. Caecum was then exteriorized, on a sterile compress soaked with 0.9% NaCl solution, and slightly damaged using a 26G needle. Grafts were then apposed on the damaged region, to promote tumor attachment, and were maintained on it using 7.0 absorbable suture filament. Caecum was then replaced in the abdomen and muscles, and skin was sutured using 4.0 non-absorbable filament. Mice, randomized for treatment, were treated 3 days after graft (oxaliplatin 3 mg/kg s.c. and/or MS-275 p.o., twice a week for three weeks). Tumor progression was monitored twice per week by bioluminescence. Mice were injected with luciferin (i.p.; 150 µL/mouse of a 25 mg mL^−1^ solution) 5 min before bioluminescence acquisition, were anesthetized with isoflurane, and then the acquisition was realized on an IVIS spectrum (UMR 1240, Clermont-Ferrand, France).

### 4.2. Materials

The following drugs were used: oxaliplatin (Leancare Ltd., Flintshire, UK) and MS-275 (Selleck Chemicals, Houston TX, USA) solutions were prepared directly before experiments in 5% glucose solution for oxaliplatin and in 0.9% (*w*/*v*) NaCl solution for MS-275. Oxaliplatin (3 mg kg^−1^, i.p.) was administered twice per week for three weeks. MS-275 (15 mg kg^−1^, p.o.) was administered half an hour before each oxaliplatin injection. The administration route of oxaliplatin changed (from i.p. to s.c.) in the graft model to avoid direct contact of the drug with the tumor.

### 4.3. Evaluation of Pain Thresholds

Cold thermal nociceptive responsiveness was assessed using the cold water (10 °C) paw immersion test. The paw of the animal was immersed in the temperature-controlled water bath until withdrawal was observed (cutoff time: 30 s). Two separate withdrawal latency time determinations were averaged.

Mechanical pain hypersensitivity was assessed using a 0.6 g bending force calibrated Von Frey hair filament, applied perpendicular to the plantar surface of the hind paw until it bent. Scores were expressed, averaging results from the left and right paws, as a number of responses elicited by five consecutive stimulations by a given filament was averaged between the left and right paw.

### 4.4. Hematological Analysis

Blood samples of animals were collected intraorbitally in a heparin tube and were kept at 4 °C until the count. Full blood counts were realized at the Biological Center of the University Hospital Center of Clermont-Ferrand, France.

### 4.5. RNA Sequencing

To investigate global mRNA changes, we used RNA sequencing analysis on L4-L5-L6 DRGs from mice treated with oxaliplatin (3 mg kg^−1^, i.p.) or vehicle twice per week for 3 weeks. At the end of the experiment (D21), mice were terminally anesthetized and L4-L5-L6 DRGs were quickly harvested, snap frozen in liquid nitrogen, and stored at −80 °C until use. Total RNA of L4-L5-L6 DRGs was extracted using RNeasy Micro Kit (Qiagen, Hilden, Germany) according to the manufacturer’s protocol. Thereafter, rRNA was removed using Illumina^®^ Ribo-Zero Plus rRNA Depletion Kit (Epicentre, Illumina, San Diego, CA, USA). RNA integrity and concentration were obtained using 2100 Bioanalyzer instrument (Agilent Technologies, Santa Clara, CA, USA) and were sent to Fasteris (https://www.fasteris.com (accessed on 6 September 2017)) for RNA sequencing. Libraries were prepared using the Illumina TruSeq stranded protocol, and an SBS-based sequencing was achieved using HiSeq 2500 platform (Illumina). Analysis was performed in different steps: splice junction mapping (TopHat2), counting (HTSeq-count), filtering, normalization (edgeR, DESeq and DESeq2) and differential analysis (edgeR, DESeq and DESeq2) were performed by Benjamin Bertin and Yoan Renaud (GreD, Clermont-Ferrand, France).

### 4.6. Bioinformatics Analysis

Quality control of sequencing was evaluated using FastQC software. High-quality reads were mapped to the Mus musculus mm9 reference genome using bowtie2 with default parameters [65]. Reads per gene were counted using HTSeq-count [66].

Normalization and differential gene expression analysis were performed using DESeq2 [67] from the SARTools package [68]. Only genes with an adjust *p*-value of <0.05 were considered as differentially expressed between the two conditions. GO and KEGG enrichment analyses were performed using the R package clusterProfiler from Bioconductor [69].

Significantly enriched GO terms were selected, respectively, according to a p value < 0.01. The gene ontology study was performed using R package ClusterProfiler (v 4.2.0). Volcano plot was generated using R package ggplot2 (v 3.3.5). Cell Type enrichment analysis (CTen) was performed using http://sbi.jp/influenza-x/cten (accessed on 6 September 2017) web application.

### 4.7. Cell Culture

The CT26 (ATCC, CRL-2638) cell line was purchased from LGC Standards (France), HT29 (ATCC, HTB-38) and T84 (ATCC, CCL-248) cell lines were a kind gift from Dr. Mathilde Bonnet (UMR 1071 Inserm, Clermont-Ferrand, France). Cell lines were grown in DMEM (CT26, HT29) or DMEM/F-12 (T84) complete medium. Cells were maintained in a humidified atmosphere at 37 °C and 5% CO2. All drugs tested in vitro were dissolved in 100% dimethyl sulfoxide (DMSO, Sigma-Aldrich, Saint Quentin Fallavier, France) solution.

### 4.8. Cell Viability Analysis

Cell viability was assessed using 3-(4,5-Dimethylthiazol-2-yl)-2,5-diphenyltetrazolium bromide (MTT) (Life Technologies, France), according to the manufacturer’s instructions. Briefly, 7500 cells were plated in a 96-well plate and treated for 48 h with oxaliplatin and/or MS-275. The optical density was read at 562 nm using an Epoch microplate spectrophotometer (BioTek, Winooski, FL, USA).

### 4.9. Apoptosis Analysis

Cells were seeded at 225 × 10^3^ cells in a 6-well plate and were treated for 48 h with oxaliplatin and/or MS-275. Briefly, the medium was removed and kept, and the cells were rinsed with PBS, trypsinized, and added to the medium. After centrifugation (10 min, 200 g), cells were homogenized in 100 µL of Binding Buffer and were stained with 5 µL propidium iodide (PI) and 1 µL Annexin V-FITC (Annexin V-FITC kit Beckman Coulter, Indianapolis, IN, USA) for 15 min at 4 °C in the dark. Afterward, 400µL of binding buffer was added, and apoptosis was analyzed by flow cytometry at CICS (Centre Imagerie Cellulaire Santé) Core Facility (Clermont-Auvergne University, France) using the LSR II flow cytometer (BD Biosciences, San Jose, CA, USA). Quantification was performed with the BD FACS Diva software version 6.1.3.

### 4.10. Cell Cycle Analysis

Cells were plated and treated as described above. Cells were collected, washed in ice-cold PBS, and fixed in 70% ice-cold ethanol for 30 min at 4 °C. After fixation, cells were washed with ice-cold PBS and were pelleted. Cells were resuspended with 1 mL of RNase A-PI solution (500 and 50 µg/mL, separately) and were stained for 30 min at 4 °C in the dark. Cell cycle analysis was performed as described above, except for the quantification that was performed with the ModFit LT software (Verity Software House, Topsham, ME, USA) version 3.0.

### 4.11. Statistical Analysis

Results were expressed as mean ± standard error of the mean. Data were analyzed using GraphPad Prism software (GraphPad software, La Jolla, CA, USA) version 7.0. The specific tests used are indicated within the text of the figure legends, and power analysis for all the tests is provided as a Appendix A.

## 5. Conclusions

The well-tolerated epigenetic drug MS-275 was tested in phase III clinical trials for the treatment of various tumor entities [70]. Our results strongly suggest the interest for combining oxaliplatin and MS-275 administration in CRC patients in order to potentiate the antiproliferative action of the chemotherapy while minimizing the neurotoxic effects of the platinum drug.

## Figures and Tables

**Figure 1 ijms-23-00098-f001:**
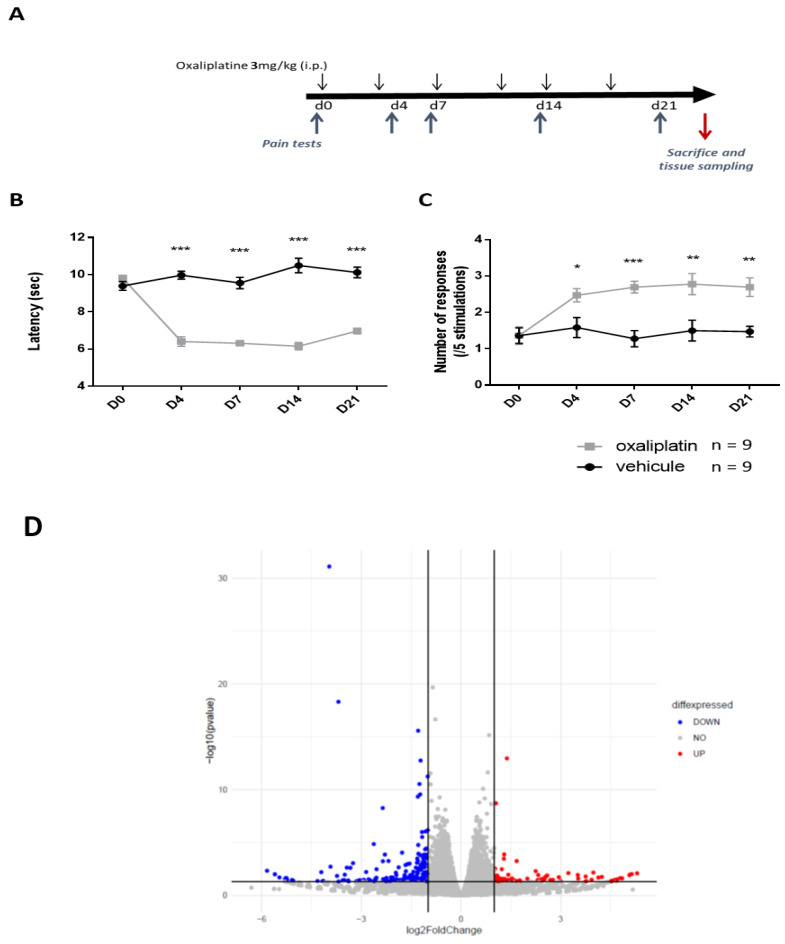
Analysis of DEG in mouse lumbar dorsal root ganglia L4 to L6 following repeated oxaliplatin administration. (**A**) Experimental design: C57Bl/6J mice received oxaliplatin (3 mg/kg, i.p.) or vehicle (Glucose 5%) twice a week for 3 weeks. Both thermal cold (**B**) and mechanical (**C**) pain hypersensitivity developed as soon as after day 4 in oxaliplatin treated group. Values are mean ± SEM (n = nine per group). Statistical analysis was performed using a two-way repeated measure analysis of variance (RM ANOVA), detailed in Appendix A, and a Tukey’s multi-comparisons post hoc test; *, *p* < 0.05, **, *p* < 0.01, ***, *p* < 0.001, vehicle *versus* oxaliplatin. (**D**). Volcano plot of all DEGs following oxaliplatin administration showing the most highly upregulated (log2fold change > 1) or downregulated (log2fold change < −1) genes. Only genes with Padj < 0.05 and log2fold change greater than 1 or smaller than −1 were used for further analysis. (**E**) Oxaliplatin administration caused the upregulation of 342 DEGs and the downregulation of 399 DEGs.

**Figure 2 ijms-23-00098-f002:**
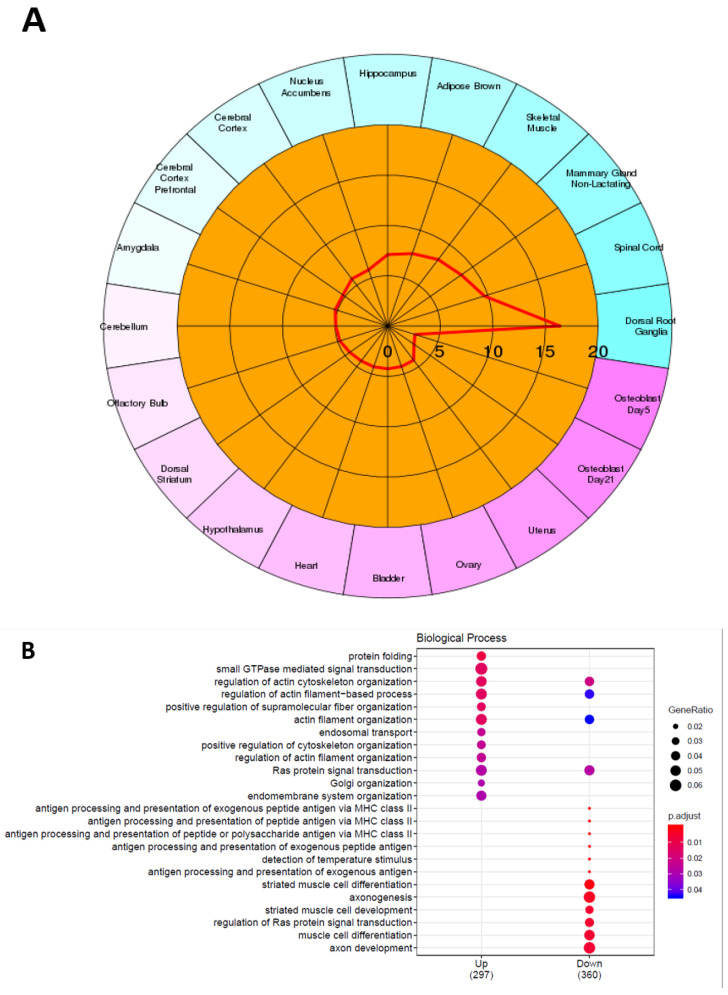
Cell type specific effect of oxaliplatin and enrichment analysis of the overlapping DEGs. (**A**) Cell type enrichment analysis (CTEN) of DEG in lumbar dorsal root ganglia L3 to L5 following repeated oxaliplatin administration. CTEN was performed using the CTen tool. The score is generated using one-sided, Fisher’s exact test for enrichment and it is shown as the -log10 of the Benjamini-Hochberg (BH) adjusted P values. Scores > 20 optimally minimize the false positive rate. (**B**,**C**) illustrate the GO enrichment analysis results: (**B**) biological process, (**C**) cellular components.

**Figure 3 ijms-23-00098-f003:**
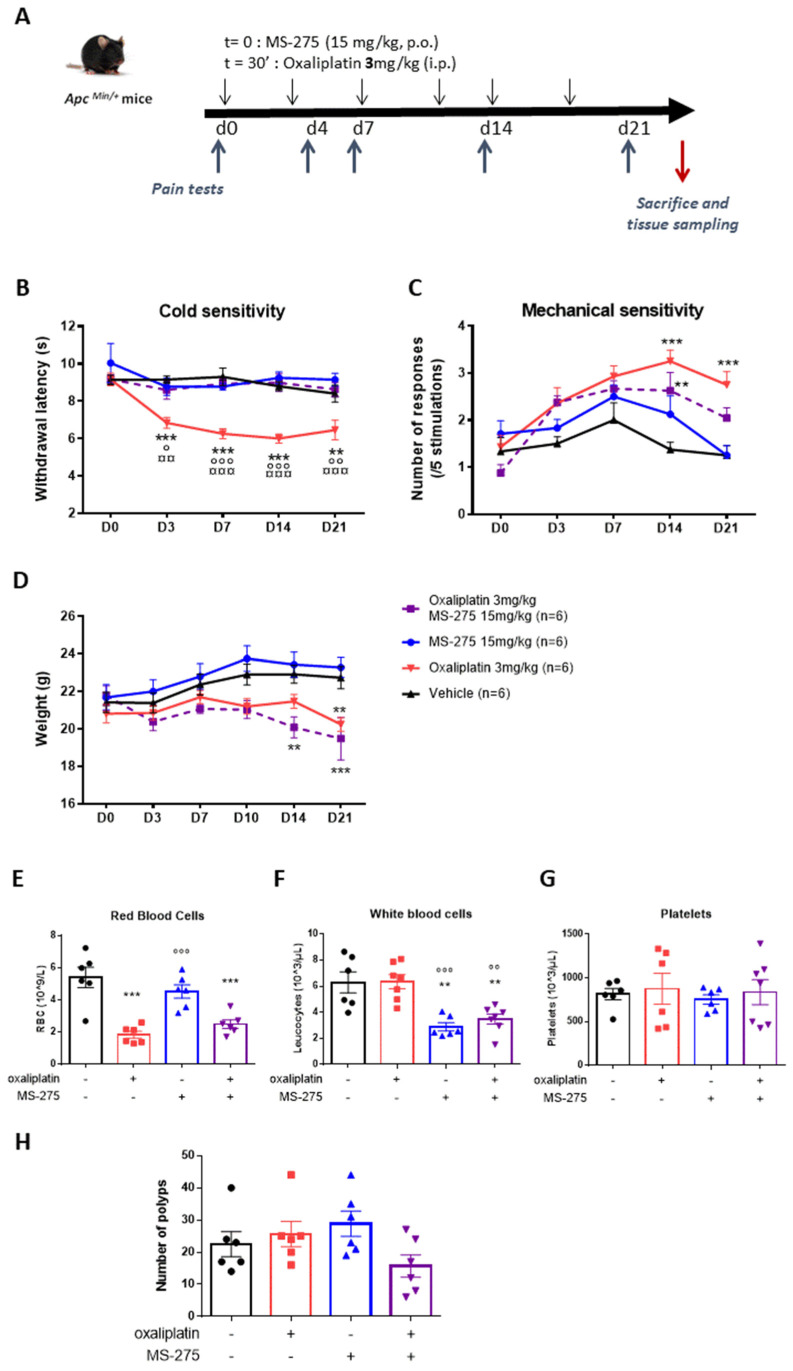
MS-275 prevents oxaliplatin-induced chronic neuropathic pain in APC^Min/+^ mice. (**A**) Experimental design: *APC^Min/+^* mice received oxaliplatin (3 mg/kg, i.p.) or vehicle (Glucose 5%) twice a week for 3 weeks. MS-275 (15 mg/kg, p.o.) was administered half an hour before each oxaliplatin injection. MS-275 significantly prevented cold (**B**) and mechanical (**C**) pain hypersensitivity induced by oxaliplatin administration. Values are mean ± SEM (n = 5/7 per group). Statistical analysis was performed using a two-way repeated measure analysis of variance (RM ANOVA), detailed in Appendix A, and a Tukey’s multi-comparisons post hoc test; **, *p* < 0.01, ***, *p* < 0.001, vehicle versus oxaliplatin; ¤¤, *p* < 0.01, ¤¤¤, *p* < 0.001, oxaliplatin versus MS-275; °, *p* < 0.05, °°, *p* < 0.01, °°°, *p* < 0.001, oxaliplatin versus oxaliplatin + MS-275. General and hematologic toxicity profile of oxaliplatin and MS-275 combination. The body weight of animals was monitored twice a week during all the experimental procedure (**D**). Effect of oxaliplatin and/or MS-275 treatment on hematological parameters, red blood counts (**E**), white blood counts (**F**) and platelets counts (**G**) was done at end point (D21). Number of polyps in *APC^Min/+^* at end-point (**H**). Values are mean ± SEM (n = 6/7 per group). Statistical analysis was performed using a two-way analysis of variance, detailed in Appendix A, and a Tukey’s multi-comparisons post hoc test; **, *p* < 0.01, ***, *p* < 0.001, versus the vehicle group; ° *p* < 0.05, °°, *p* < 0.01, °°°, *p* < 0.001, versus the oxaliplatin group.

**Figure 4 ijms-23-00098-f004:**
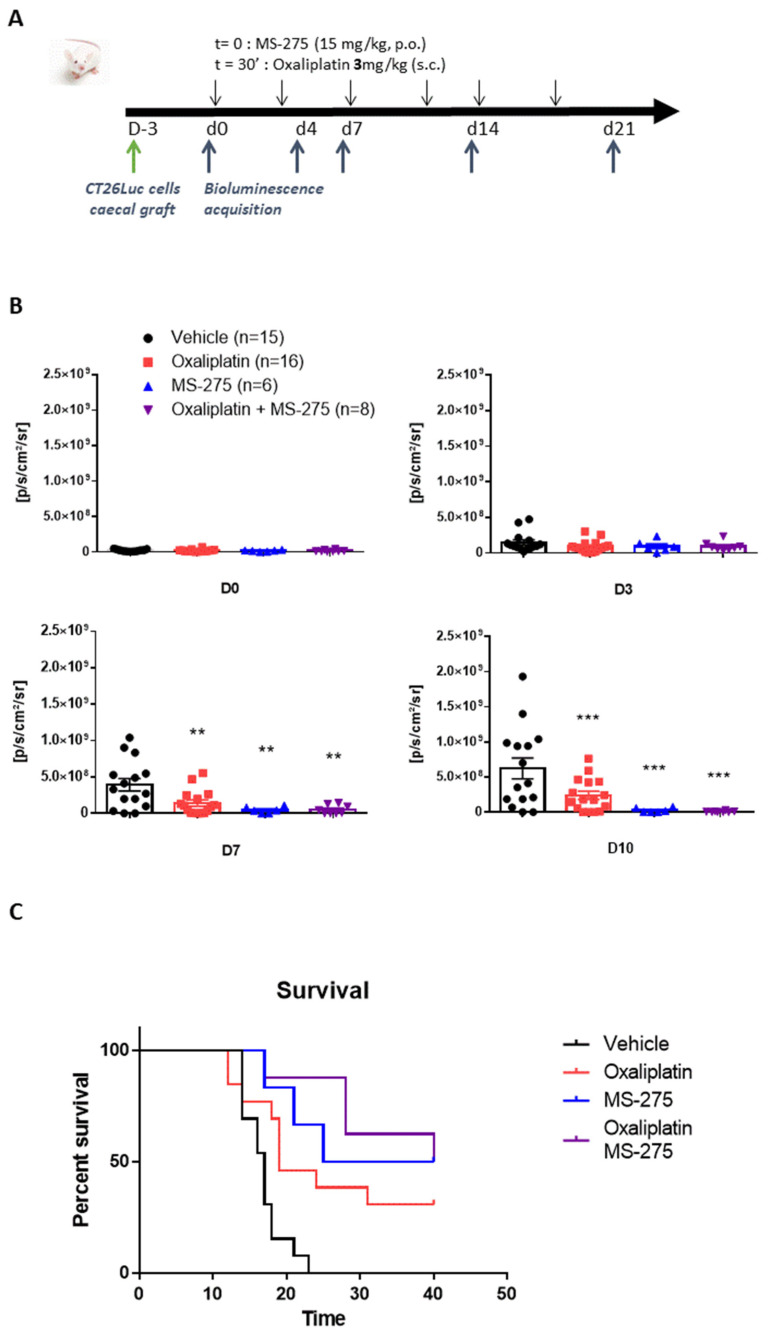
Oxaliplatin and MS-275 exert synergic antitumoral effect in a mouse model of orthotopic CRC. (**A**) Experimental design: BALB/c AnN mice, randomized for treatment, were treated 3 days after graft with oxaliplatin (3 mg/kg i.p.) and/or MS-275 (15 mg/kg, p.o.), twice a week for three weeks. MS-275 was administered half an hour before each oxaliplatin injection. (**B**) MS-275, either alone or combined with oxaliplatin strongly slowed cancer progression in this model. Bioluminescence results are shown until D10 because of the occurrence of death events at D12 in the vehicle group in which 100% animals died before D23 (Figure 4C). Statistical analysis was performed using a two-way analysis of variance, detailed in Appendix A, and a Tukey’s multi-comparisons post hoc test: **, *p* < 0.01, ***, *p* < 0.001, *versus* the vehicle group. (**C**) Oxaliplatin and MS-275 improves the survival of mice, and drugs association demonstrated an increased benefit for survival as shown by their synergistic effect on mean and median survival (Appendix A). Statistical analysis was performed using Kruskal–Wallis test, detailed in Appendix A, and a Dunn’s multi-comparisons post hoc test; **, *p* < 0.01, versus vehicle.

**Figure 5 ijms-23-00098-f005:**
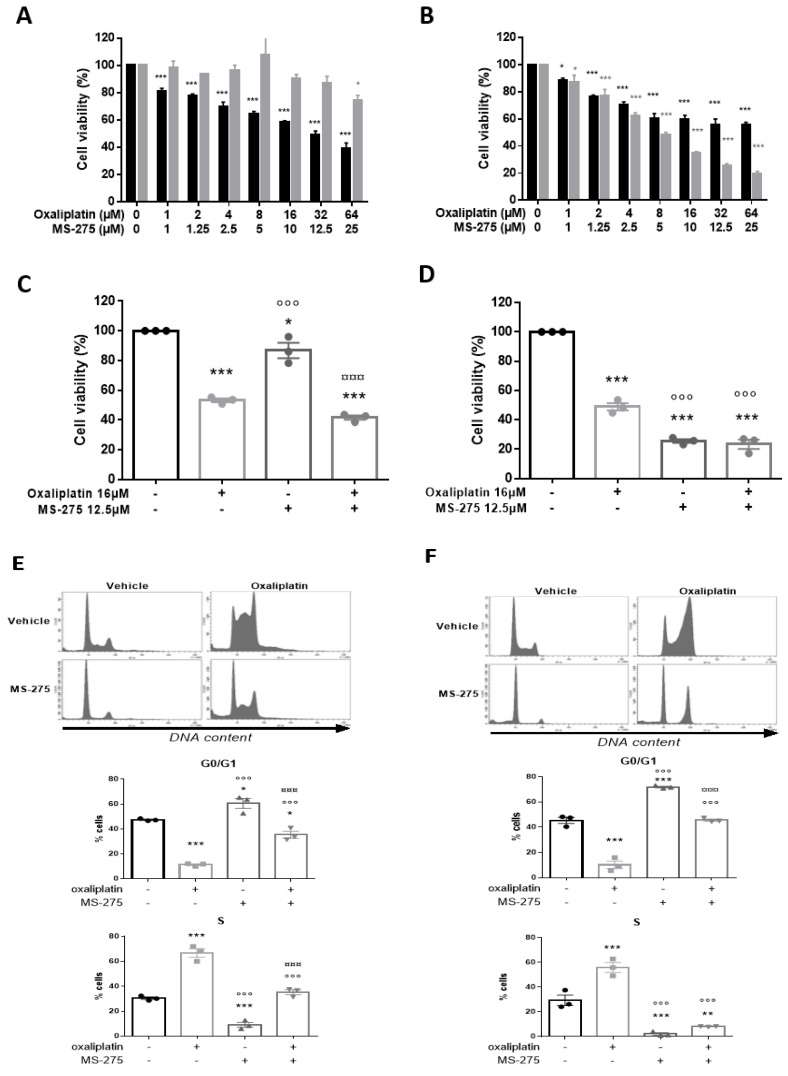
Oxaliplatin and MS-275 effects on in human colon cancer cell lines viability and cell cycle. Oxaliplatin dose-dependently inhibited survival of T84 (**A**) and HT29 (**B**) colorectal cancer cells as measured by inhibition of mitochondrial dehydrogenase activity (MTT assay). MS-275 dose-dependently inhibited survival HT29 cells (**B**) while significantly decreasing T84 cell viability only at the highest dose tested (**B**). Combined effect of oxaliplatin (16 µM) and MS-275 (2.5 µM) on T84 (**C**) and on HT29 (**D**) cell viability and on T84 (**E**) and HT29 (**F**) cells cycles. The average results from at least three independent experiments are presented. Values are expressed as mean ± SEM. Statistical analysis was performed using two-way ANOVA, detailed in Appendix A, with Tukey’s multi-comparisons post-hoc test; * for *p* < 0.05; ** *p* < 0.01, *** *p* < 0.001, versus the vehicle group; ° for *p* < 0.05, °°° for *p* < 0.001, versus the oxaliplatin group; ¤¤ for *p* < 0.01, ¤¤¤ for *p* < 0.001, versus the MS-275 group.

**Figure 6 ijms-23-00098-f006:**
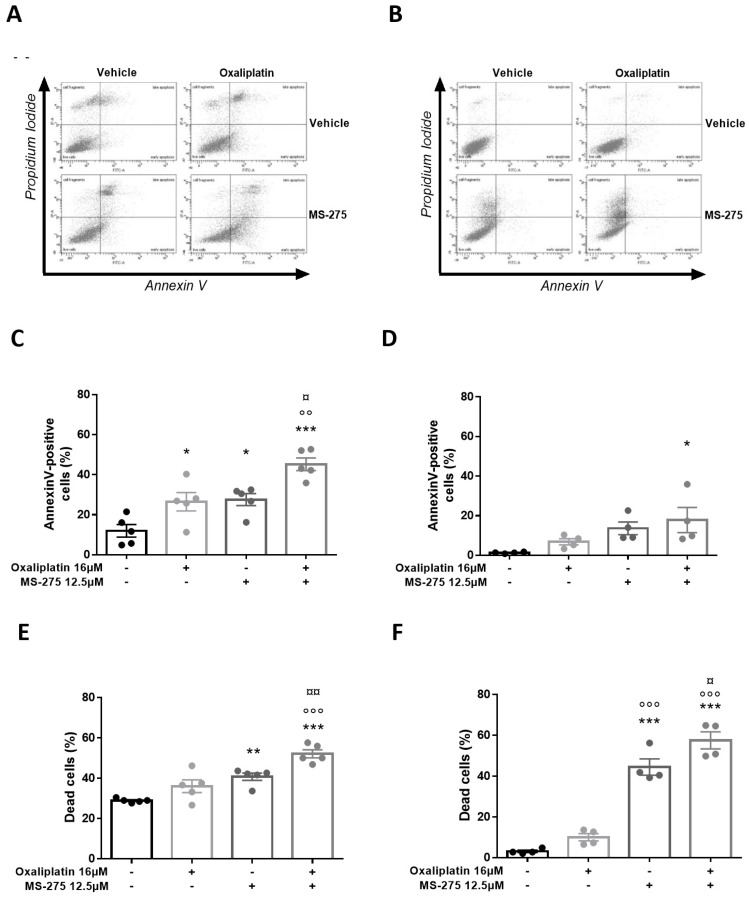
Oxaliplatin and MS-275 effects on apoptosis and cell death in T84 and HT29 cells. Annexin V and Propidium Iodide (PI) staining was performed on T84 (**A**) and HT29 (**D**) cell lines followed by flow cytometry analysis. Effects of oxaliplatin (16 µM) with or without MS-275 (2.5 µM) on T84 apoptosis (**B**) and cell death (**C**) and on HT29 apoptosis (**E**) and cell death (**F**). The average results from at least three independent experiments are presented. Values are expressed as mean ± SEM. Statistical analysis was performed using two-way ANOVA, detailed in Appendix A, with Tukey’s multi-comparisons post-hoc test; * for *p* < 0.05; ** for 0.01; *** *p* < 0.001, versus the vehicle group; °° for *p* < 0.01, °°° for *p* < 0.001, versus the oxaliplatin group; ¤ for *p* < 0.05, ¤¤ for *p* < 0.01, versus the MS-275 group.

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
