# Peer review of "The Class I HDAC Inhibitor, MS-275, Prevents Oxaliplatin-Induced Chronic Neuropathy and Potentiates Its Antiproliferative Activity in Mice"

_ijms, 2021, doi:10.3390/ijms23010098_

Round 1

Reviewer 1 Report

This is an interesting study as authors show that their drug (MS-275) can prevent the inducement of oxaliplain-induced neuropathic pain, which is clinically an important issue. Moreover, several data support the effect. However, some explanations are not clear and more detailed explanation are needed to help the readers clearly understand the effect of MS275 and its mechanism of action.  

1) What could be the relation between the analgesic effect of MS-275 and the results obtained from the analysis of DEG? Because, the authors mention that class I HDACs were not altered when measured at day 21. Is it that MS-275 prevented the development of pain through K+ channel regulation in the beginning of the pain and that this effect last till day 21?

2) On Figure 3, MS-275 is less effective (or not) against mechanical allodynia. What could be the reason? Could it be due to the difference in the peripheral nerve type? Also, the gene differences could be applied to both cold and mechanical allodynia?

3) Figure 3D, MS-275 could prevent the pain development and decrease the tumor by having additive effect with oxaliplatin. Then, why the body weight decreased similarly to oxaliplatin treated mice? please explain.

4) It is interesting that the leukocytes numbers did not change in oxaliplatin treated mice compared to control. Also, MS-275 significantly decreased the leukocytes numbers. In chemotherapy treated patients the white blood cells count is an important marker for the continuation of the treatment. Can this be a problem? 

Author Response

Point by point answer to reviewer#1:

This is an interesting study as authors show that their drug (MS-275) can prevent the inducement of oxaliplain-induced neuropathic pain, which is clinically an important issue. Moreover, several data support the effect. However, some explanations are not clear and more detailed explanation are needed to help the readers clearly understand the effect of MS275 and its mechanism of action. 

  • What could be the relation between the analgesic effect of MS-275 and the results obtained from the analysis of DEG? Because, the authors mention that class I HDACs were not altered when measured at day 21. Is it that MS-275 prevented the development of pain through K+ channel regulation in the beginning of the pain and that this effect last till day 21?

As mentioned page 17 lines 318-320: “Since MS-275 significantly prevented oxaliplatin-induced acute neuropathy [51], we hypothesized that this molecule could also act as a preventive therapeutic drug in a “programmed” chronic oxaliplatin-induced neuropathic condition.”, we effectively hypothesized that since MS-275 prevented the development of pain through K+ channel regulation in the beginning of the pain, this effect should last till day 21. Another possibility would be that central (spinal and/or supraspinal) epigenetics involving HDAC overexpression are involved in central sensitization induced by repeated oxaliplatin administration and consequently MS275 might exert its beneficial effect at the central level. This hypothesis is mentioned page 17 lines 327-328.

  • On Figure 3, MS-275 is less effective (or not) against mechanical allodynia. What could be the reason? Could it be due to the difference in the peripheral nerve type? Also, the gene differences could be applied to both cold and mechanical allodynia?

We are grateful to Reviewer #1 for his/her suggestions. Indeed, even if the ANOVA states for a high significant effect of treatment on mechanical hypersensitivity, MS-275 is less effective against oxaliplatin-induced mechanical hypersentivity as compared with cold hypersensitivity. On the contrary, we previously observed a similar benefic effect of MS-275 on acute oxaliplatin-induced cold and mechanical hypersensitivity. Thus, we think that these differences may not be due to the difference in the peripheral nerve type and that the gene differences could be applied to both cold and mechanical allodynia. The present result would need further investigation.

  • Figure 3D, MS-275 could prevent the pain development and decrease the tumor by having additive effect with oxaliplatin. Then, why the body weight decreased similarly to oxaliplatin treated mice? please explain.

We are not sure to get the question. We do not think that there is direct relationship between the preventive effect of MS-275 on oxaliplatin-induced pain hypersensitivity and polypes development from the one side and body weight gain from the other side.  Oxaliplatin toxicity is similar between oxaliplatin- and oxaliplatin + MS275- treated animals when we look both at the body weight and red blood cells counts. Thus, MS-275 could neither prevent erythropenia nor body weight slow down. As mentioned page 11 lines 189-191, these results suggest that co-administration of MS-275 and oxaliplatin would have additive, but no synergic, hematological toxicities that should be taken into consideration.

  • It is interesting that the leukocytes numbers did not change in oxaliplatin treated mice compared to control. Also, MS-275 significantly decreased the leukocytes numbers. In chemotherapy treated patients the white blood cells count is an important marker for the continuation of the treatment. Can this be a problem?

Reviewer#1 is right. We now more clearly mention this point (page 11 lines 191-192) as follows: these results suggest that co-administration of MS-275 and oxaliplatin would have additive, but no synergic, hematological toxicities. This should be taken into consideration since in chemotherapy treated patients the white blood cells count is an important marker for the continuation of the treatment.

Reviewer 2 Report

Lamoine et al. analyzed the interaction of oxaliplatin and MS-275 in mice and CRC tumor cells.

Critique:

Is oxaliplatin really the 1st line therapy? what about 5-FU, especially in relation to side effects?

Which measures were taken to ensure animal welfare in pain experiments?

Quality of figures is suboptimal regarding solution and readability.

Avoid French words, vehicle not vehicule.

Fig. 2A: This experiment and its implications are not explained well.

Global analyses: Is it possible that no alteration in DNA replication stress/DNA damage markers were seen in oxaliplatin-treated cells? This needs an explanation.

Fig. 3H: Not significant?

Fig. 5: What are the concentrations of the drugs? Not stated.

In sum, the manuscript deals with an interesting and relevant subject. I have some serious concerns. (i) The quality of the figures must be improved (resolution, fonds,…). (ii) The different effects of oxaliplatin and MS-275 on normal and transformed cells must be discussed. Effects of MS-275 that are known from global analyses should be discussed in this context, see for example, Kiweler et al., Arch Toxicol. 2018 Jul;92(7):2227-2243. (iii) The concentration of MS-275 is far too high to show physiologic relevance. What is realistic in humans? See recent publication by Nguyen et al., Cells. 2021 Sep 23;10(10):2520. Same for oxaliplatin, what is achievable in humans? See for example, Rauch et al., Oncotarget 2018. Therefore, I suggest pointing this out and to repeat some of the cell culture experiments with 1-2 µM MS-275 and 5 µM oxaliplatin. (iv) What is known about APC and HDAC2, see Zhu et al., Cell Cycle. 2004 Oct;3(10):1240-2. How can this effect the data from the different mouse models? (v) More discussion is needed regarding published literature on the impact of MS-275 on CRC cells.

Author Response

Point by point answer to reviewer#2:

Lamoine et al. analyzed the interaction of oxaliplatin and MS-275 in mice and CRC tumor cells.

 Critique:

 Is oxaliplatin really the 1st line therapy? what about 5-FU, especially in relation to side effects?

Oxaliplatin and irinotecan in combination with 5-fluorouracil (5-FU) are standard treatment options for primary and metastasized colorectal cancer (Carrato 2008). Oxaliplatin is mostly administered in combination with 5-FU and folinic acid. However, no neurotoxic effect of 5-FU has been reported to date. The neurotoxicity is known to be uniquely due to oxaliplatin (Pachman et al., J Clin Oncol 2015). Consistently, we previously used such combination in mice and shown that there was no additive adverse effect than those known to be induced by oxaliplatin alone (Poupon et al., 2018). We made this point clearer in the introduction page 5 lines 58-60: “Oxaliplatin, in combination with 5-fluorouracil, is a standard treatment option for primary and metastasized colorectal cancer [8] that induces a peripheral neuropathy, known to be uniquely due to oxaliplatin [40], which can lead to dose limitation and treatment discontinuation [3; 40].”  

Which measures were taken to ensure animal welfare in pain experiments?

All efforts were taken at each stage of the experiments to limit the numbers of animals used and any discomfort to which they might be exposed, especially in pain experiments. This now appears in the methods section page 21 lines 416-418.

Quality of figures is suboptimal regarding solution and readability.

The quality of figures was improved and colors were added to make it more clear for the readers.

Avoid French words, vehicle not vehicule.

This has been corrected. Moreover, we use MDPI service for english editing of the manuscript.

 Fig. 2A: This experiment and its implications are not explained well.

As the DRGs are a heterogeneous tissue containing the cell bodies of peripheral nerves, epithelial cells, fibroblasts, glial cells, and immune cells, DEGs discovered following treatment with oxaliplatin cannot be unequivocally assigned to sensory neurons alone. We thus performed CTEN, a web-based analytical platform using a highly expressed, cell-specific gene database to identify enriched cell types from transcriptomic data [48] to gain insights into cell-specific effects of oxaliplatin. This now appears page 9, lines 135-140.

Global analyses: Is it possible that no alteration in DNA replication stress/DNA damage markers were seen in oxaliplatin-treated cells? This needs an explanation.

We now discussed this point page 20, lines 390-394.  

Fig. 3H: Not significant?

As mentioned page 11, line 200 to page 12 line 204: MS-275 failed to significantly decrease the number of polyps in APCMin/+ mice, but a significant interaction was detected using the two-way ANOVA analysis (two-way ANOVA analysis: oxaliplatin effect (P = 0.2037); MS-275 effect (P = 0.6351), interaction (P = 0.0443)), suggesting a synergistic effect of combined drugs treatment in this model.

Fig. 5: What are the concentrations of the drugs? Not stated.

This has been corrected.

In sum, the manuscript deals with an interesting and relevant subject. I have some serious concerns. (i) The quality of the figures must be improved (resolution, fonds,…).

We have improved the quality of the figures and included colors to make it more clear to the readers. .

 (ii) The different effects of oxaliplatin and MS-275 on normal and transformed cells must be discussed. Effects of MS-275 that are known from global analyses should be discussed in this context, see for example, Kiweler et al., Arch Toxicol. 2018 Jul;92(7):2227-2243.

These points are now discussed on page 20, lines 390-394: HDACi are epigenetic drugs that enhance protein acetylation and thereby influence a large number of cellular functions. Global assays and analyses revealed that HDACi promote an induction of replicative stress and DNA damage [35], disrupt functional EMT/MET protein expression signatures and trigger apoptosis of cultured cancer cells [34].

(iii) The concentration of MS-275 is far too high to show physiologic relevance. What is realistic in humans? See recent publication by Nguyen et al., Cells. 2021 Sep 23;10(10):2520. Same for oxaliplatin, what is achievable in humans? See for example, Rauch et al., Oncotarget 2018. Therefore, I suggest pointing this out and to repeat some of the cell culture experiments with 1-2 µM MS-275 and 5 µM oxaliplatin.

A recent report, cited by Kiweler et al. (2020), found large variation of maximal plasma concentrations after MS-275 administration from 4 to 53.1 ± 92.4µM in humans (Connolly et al. 2017; Kurmasheva et al. 2019), whereas therapeutically achievable concentrations of oxaliplatin has been reported to be around  5-10 µM (Ehrsson et al., 2002). We effectively show results obtained with combination of oxaliplatin (16µM) and MS-275 (12.5µM) corresponding to concentrations in the upper range, but still these are therapeutically achievable concentrations for both drugs. We now point this out on page 13 lines 228-233. Moreover, as suggested by reviewer#2, we also replaced data obtained with oxaliplatin (16µM) and MS-275 (12.5µM) in figures 5 and 6 by data obtained with oxaliplatin (16µM) and MS-275 (2.5µM). Apoptosis and cell death effect obtained with oxaliplatin (16µM) and MS-275 (12.52µM) are now mentioned in a Supplementary Figure 1.

(iv) What is known about APC and HDAC2, see Zhu et al., Cell Cycle. 2004 Oct;3(10):1240-2. How can this effect the data from the different mouse models?

We thank Reviewer#2 to highlight this point. Indeed, it is known that loss of APC function leads to c-Myc-dependently increased expression of HDAC2 in intestinal epithelial cells and tumors (Zhu et al., Cancer Cell 2004). Inhibition of the enzymatic activity of class I HDACs and induction of HDAC2 degradation with valproic acid has previously been shown to reduce the number and sizes of intestinal adenomas in APCMin/+ mice (Zhu et al., Cancer Res 2004). We hypothesized that the lack of effect of MS-275 in our conditions could rely on the dose used and/or on the different inhibitory profile of valproic acid and MS-275 on the different HDAC isoforms. This now appears on page 18, lines 350-356. 

(v) More discussion is needed regarding published literature on the impact of MS-275 on CRC cells.

We added several points in the discussion on page 20, lines 396-404: It is increasingly recognized that HDACi sensitize tumor cells to chemotherapeutics causing replication stress and DNA damage [8–10]. Combined application of irinotecan and entinostat synergistically kills CRC cells in vitro and in vivo and the induction of Acetyl-p53-BAK complexes induce MOMP upon RS C-terminal hyperacetylation of p53 is an absolute requirement for apoptosis induction by irinotecan and entinostat (Marx et al., 2021).

Round 2

Reviewer 2 Report

Authors have significantly improved the manuscript and have satisfactorily addressed all comments. 

Author Response

We thank again Reviewer"2 for his/her comments and very constructive suggestions.